# Relationships of Homophobic Bullying Victimization during Childhood with Borderline Personality Disorder Symptoms in Early Adulthood among Gay and Bisexual Men: Mediating Effect of Depressive Symptoms and Moderating Effect of Family Support

**DOI:** 10.3390/ijerph19084789

**Published:** 2022-04-14

**Authors:** Huang-Chi Lin, Yu-Ping Chang, Yi-Lung Chen, Cheng-Fang Yen

**Affiliations:** 1Department of Psychiatry, School of Medicine, College of Medicine, Kaohsiung Medical University, Kaohsiung 80708, Taiwan; cochigi@kmu.edu.tw; 2Department of Psychiatry, Kaohsiung Medical University Hospital, Kaohsiung 80756, Taiwan; 3School of Nursing, The State University of New York, University at Buffalo, New York, NY 14214-3079, USA; yc73@buffalo.edu; 4Department of Healthcare Administration, College of Medical and Health Science, Asia University, Taichung 41354, Taiwan; 5Department of Psychology, College of Medical and Health Science, Asia University, Taichung 41354, Taiwan; 6College of Professional Studies, National Pingtung University of Science and Technology, Pingtung 91201, Taiwan

**Keywords:** gay and bisexual men, homophobic bullying, borderline personality disorder, depression, psychological well-being, family support

## Abstract

This study investigated (1) the associations of homophobic bullying victimization in childhood with borderline personality disorder (BPD) symptoms in early adulthood among gay and bisexual men; (2) the mediating effect of depressive symptoms on the association between homophobic bullying victimization and BPD symptoms, and (3) the moderating effects of perceived family support on the association between homophobic bullying victimization and BPD symptoms. A total of 500 gay or bisexual men aged between 20 and 25 years were recruited into this study. The experiences of physical, verbal and social relationship bullying victimization during childhood were evaluated using the Mandarin Chinese version of the School Bullying Experience Questionnaire. The experiences of cyberbullying victimization during childhood were evaluated using the Cyberbullying Experiences Questionnaire. BPD symptoms were assessed using the Borderline Symptom List. Depressive symptoms were examined using the Center for Epidemiologic Studies Depression Scale. Perceived family support was evaluated using the Family Adaptation, Partnership, Growth, Affection, and Resolve index. The results of mediation analyses demonstrated that all the types of homophobic bullying victimization in childhood were directly associated with BDP symptoms in young adulthood as well as indirectly associated with BPD symptoms through the mediation of depressive symptoms. The results of moderation analyses demonstrated that the association between homophobic bullying victimization and BPD symptoms decreased when the individuals had more family support. Intervention programs to reduce homophobic bullying victimization and enhance family support for gay and bisexual men and their families are necessary. Interventions to improve depressive and BPD symptoms among gay and bisexual men are also necessary, especially for those who experienced homophobic bullying victimization during childhood.

## 1. Introduction

### 1.1. Homophobic Bullying Victimization and Borderline Personality Disorder Symptoms in Lesbian, Gay and Bisexual Individuals

Lesbian, gay and bisexual (LGB) individuals may experience multiple forms of homophobic bullying victimization during childhood [1,2,3]. The experiences of homophobic bullying victimization can adversely affect victims’ mental and physical health, self-concept, and interpersonal relationships throughout their lives [4,5,6]. The negative effect of homophobic bullying victimization on personality development is an important issue for LGB individuals. Research found that LGB individuals were more likely to be diagnosed with borderline personality disorder (BPD) than heterosexual individuals [7]. In epidemiological studies of adults in the USA, prevalence for BPD was between 0.5% and 5.9% in the general US population [8,9] with a median prevalence of 1.35% [10]. Previous studies around the world, most conducted in the USA and Europe, found that in clinical populations, BPD is the most common personality disorder, with a prevalence of 10% of all psychiatric outpatients and between 15% and 25% of inpatients [11,12].

The cardinal symptoms of BPD include emotional instability, feelings of emptiness, and impulsive behaviors [13] and can result in psychosocial dysfunction, particularly suicide [13,14,15]. Interactions between biological factors (e.g., genetic vulnerability and temperament) and adverse childhood events (e.g., physical or sexual abuse) contribute to the development of BPD [16,17,18]. Prospective studies have identified bullying victimization as an adverse event experienced during childhood that may increase the risk of BPD symptoms later in life [19,20,21]. Furthermore, the impulsive and dysregulated behaviors of children with BPD traits may elicit bullying from peers [22]; bullying victimization, in turn, can subsequently increase the risk of BPD [23].

Studies have reported many of those diagnosed as having BPD identified as gay or bisexual [24,25]. For example, a clinical study found that homosexuality was ten times more common in men and six times more common in women with BPD compared to both the general population and a depressed control group [26]. A ten-year longitudinal study on a clinical sample found that approximately one-third of both men and women with BPD engaged in homosexual relationships [27]. A study on the individuals referred for psychiatric treatment revealed that LGB individuals were more likely to be diagnosed with BPD than heterosexual individuals (odds ratio = 2.43, *p* < 0.001) [7].

However, some studies have indicated that clinicians are predisposed to providing a diagnosis of BPD to LGB individuals [7,28]. Rodriguez-Seijas et al. [7] hypothesized that stigma and discrimination may lead to sexual minority stress and increase victims’ interpersonal difficulties, rejection sensitivity, impulsive behaviors, suicidality, and self-harm; these stress-related symptoms are similar to BPD symptoms. However, whether homophobic bullying victimization in childhood increases the risk of self-reported BPD symptoms in young adulthood among gay and bisexual men has not yet been well examined. Moreover, LGB individuals may experience multiple types of bullying such as physical, verbal, social relationship, and cyber bullying; the effects of various types of homophobic bullying victimization should be considered. A literature review also reported that gender role nonconformity significantly increases the risk of experiencing sexuality-related bullying in both heterosexual and LGB populations [29]. Therefore, the experiences of both homophobic bullying victimization as a result of sexual orientation and gender nonconformity should be considered.

### 1.2. Mediating Effect of Depression

Examining the mediators and moderators of the effect of homophobic bullying victimization on BPD symptoms may provide evidence for developing intervention programs. A systematic review and meta-analysis of population-based studies reported that the prevalence rates of depressive symptoms and depressive disorder were higher in LGB adolescents than in heterosexual adolescents [30]. The most commonly cited cause of depression in LGB individuals is “minority stress”, defined as the unique stress experienced by LGB individuals living in a social environment characterized by anti-LGB, or “heterosexist”, prejudice and stigma [31]. According to minority stress theory [31], exposure to homophobic bullying is one of the distal stressors that may leading to mental health problems such as depressive symptoms among LGB individuals [32]. Moreover, depressive symptoms are prevalent among individuals with BPD [33,34]. A follow-up study observed that the development of depression mediates the effect of childhood exposure to violence on BPD symptoms in adulthood [35]. A study reported that emotion regulation difficulties caused by childhood emotional trauma mediate the association of early-life stress with an increased risk of BPD symptoms later in life [36]. However, no study has examined the mediating role of depressive symptoms in the association between homophobic bullying victimization and BPD symptoms among gay and bisexual men.

### 1.3. Moderating Effect of Familial Support

Socioecological theory identifies family as a microsystem that individuals interact with in their early lives [37]; therefore, familial contexts may play a crucial role in personality development. A study reported that family support can protect LGB youth from depression [6]. Furthermore, studies examining the general population have determined that family violence [38,39] and caregivers’ harsh punishment and lack of warmth [40] can increase the risk of BPD symptoms. However, no study has explored the moderating role of family support in the relationship between homophobic bullying victimization and BPD symptoms among gay and bisexual men.

### 1.4. Mental Health among LGB Individuals in Taiwan

The attitude toward homosexuality has even been supposed to be socially tolerable among Taiwanese society in the past two decades [41]. The Taiwan government has enacted the Sexual Harassment Prevention Act [42], Act of Gender Equality in Employment [43], and Gender Equity Education Act [44] as well as legislated the same-sex relationship [45] to protect the rights of LGB individuals from being violated. However, people in Taiwan traditionally regard homosexuality as a challenge to the family obligations mandated in Confucianism, and in particular, they require their offspring to continue the family bloodline [41]. Gay and bisexual men are still seen as the symbol of disseminating human immunodeficiency virus infection and deteriorating traditional family values [46]. Victimization of homophobic bullying [47,48], public stigma [49], microaggression [50,51], and internalized stigma [52] are prevalent among LGB individuals. Especially, the groups opposing same-sex marriage in Taiwan spread a considerable amount of incorrect information and rumors to malign LGB individuals through social media and public media; these misleading portrayals and negative stereotypes spread in the media demoralized LGB individuals and directly disturbed their emotional regulation [53,54,55,56,57,58]. LGB individuals have higher rates of suicidality [59,60], addictive substances use [61,62], problematic internet and smartphone use [63], poor quality of life [64,65], depression [66], and psychological inflexibility [67] compared with heterosexual ones in Taiwan. Therefore, mental health is an important issue warranted further study among LGB individuals in Taiwan.

### 1.5. Aims of This Study

This study investigated (1) the associations of homophobic bullying victimization in childhood with BPD symptoms in early adulthood among gay and bisexual men; (2) the mediating effect of depressive symptoms on the association between homophobic bullying victimization and BPD symptoms, and (3) the moderating effects of perceived family support on the association between homophobic bullying victimization and BPD symptoms. We hypothesized that (1) homophobic bullying victimization in childhood would be significantly associated with BPD symptoms in early adulthood, (2) depressive symptoms would mediate the association between homophobic bullying victimization and BPD symptoms, and (3) perceived family support would moderate the association between homophobic bullying victimization and BPD symptoms.

## 2. Methods

### 2.1. Participants

The methods used to recruit 500 gay and bisexual men to this study were the same as those of Wang et al. [68]. In brief, we recruited the participants by posting an advertisement on social media, including Facebook, Twitter, and LINE (a direct messaging app), the Bulletin Board System (a popular application dedicated to the sharing or exchange of messages on a network), and the homepages of three health promotion and counseling centers for lesbian, gay, and bisexual (LGB) individuals from August 2015 to July 2017. In this study, we included Taiwanese men who self-identified as homosexual or bisexual, were aged between 20 and 25 years, and lived in Taiwan. We excluded patients having any deficits in cognitive function (e.g., traumatic brain injury, intellectual disability, and severe substance use) that would impede their understanding of the study’s purpose or the questionnaires’ content. In total, 371 gay men and 129 bisexual men participated in this study and provided informed consent prior to assessment; their mean age was 22.9 (standard deviation = 1.6) years. This study was approved by the Institutional Review Board of Kaohsiung Medical University Hospital (KMUHIRB-F(I)-20150026).

### 2.2. Measures

#### 2.2.1. Bullying Victimization

We examined two types of bullying victimization experiences during childhood: enacted stigma victimization and cyberbullying victimization. The experiences of enacted stigma [69] victimization during childhood were evaluated using the Mandarin Chinese version of the School Bullying Experience Questionnaire (MC-SBEQ) [70]. The original MC-SBEQ evaluated the individuals’ victimization and perpetration of physical, verbal, and social relationship bullying (e.g., ostracizing someone; calling someone a mean nickname; speaking ill of someone; beating someone up; forcing someone to run errands; and taking away someone’s money, school supplies, or snacks). We used the victimization subscale of the MC-SBEQ to evaluate the participants’ self-reported experiences of homophobic bullying victimization as a result of their gender nonconformity (six items) and sexual orientation (six items) during childhood (e.g., “How often have others beaten you up because they thought of you as a sissy [they found you homosexual or bisexual]?”). The participants rated the items on a 4-point Likert scale from 0 (never) to 3 (all the time). A higher total score indicated more severe enacted stigma victimization. The original MC-SBEQ had acceptable internal consistency reliability (Cronbach’s α = 0.73) and 1-month test–retest reliability (intraclass correlation coefficient [ICC] = 0.78) and an adequate factor structure [70]. A previous study adapting the MC-SBEQ for measuring the victimization of bullying due to gender nonconformity and sexual orientation in gay and bisexual men and found the total McDonald’s ω values of were 0.85 and 0.92, respectively, indicating a acceptable internal consistency [47]. The Cronbach’s α values of the subscale for measuring bullying victimization as a result of gender nonconformity and sexual orientation in the present study were 0.79 and 0.82, respectively.

The experiences of cyberbullying victimization during childhood were evaluated using the Cyberbullying Experiences Questionnaire (CEQ) [71]. The original CEQ evaluated the individuals’ victimization and perpetration of cyberbullying (e.g., posting mean or hurtful comments about someone; posting upsetting pictures, photos, or videos about someone; and spreading rumors about someone through emails, blogs, or social media). The present study adopted the victimization subscale of the CEQ to examine the participants’ self-reported experiences of homophobic cyberbullying victimization because of gender nonconformity (three items) and sexual orientation (three items) during childhood (e.g., “How often have other people posted hurtful or mean comments on you through the Internet such as emails, blogs, and social media because they thought of you as a sissy [they found you homosexual or bisexual]?”). The participants rated the items on a 4-point Likert scale from 0 (never) to 3 (all the time). A higher total score indicated more severe cyberbullying victimization. The original CEQ had acceptable internal consistency reliability (Cronbach’s α = 0.70) and congruent validity with depression (Pearson’s correlation *r* = 0.42) [71]. The Cronbach’s α values of the subscale for measuring cyberbullying victimization because of gender nonconformity and sexual orientation in the present study were 0.71 and 0.86, respectively.

#### 2.2.2. BPD Symptoms

The participants’ BPD symptoms in the recent 1 week were assessed using the 23-item self-rated Mandarin Chinese version of Borderline Symptom List (MC-BSL-23, e.g., “Everything seemed senseless to me” and “I felt as if I was far away from myself”) [72]. Each item was rated on a 5-point Likert scale from 0 (not at all) to 4 (very strong). A higher total score of the MC-BSL-23 indicated more severe BPD symptoms. The original BSL-23 had high internal consistency (Cronbach’s α = 0.97) and 1-week test–retest reliability (Pearson’s correlation *r* = 0.82) and high validity to differentiate the individuals with BPD from those with a diagnosis of major depressive disorder, schizophrenia, posttraumatic stress disorder, or attention-deficit/hyperactive disorder [72]. The MC-BSL-23 had acceptable internal consistency (Cronbach’s α = 0.93), 3-month test–retest reliability (Pearson’s *r* = 0.71), and congruent validity with the Beck Depression Inventory-II [73] (Pearson’s correlation *r* = 0.72) [74]. The Cronbach’s α of the MC-BSL-23 in the present study was 0.94.

#### 2.2.3. Depressive Symptoms

The participants’ depressive symptoms in the recent 1 month were examined using the 20-item self-administered Mandarin Chinese version [75] of the Center for Epidemiologic Studies Depression Scale (MC-CES-D) (e.g., “I felt depressed” and “I felt that I could not shake off the blues even with help from my family or friends”) [76]. The participants rated each item on a 4-point Likert scale from 1 (rarely or none of the time) to 4 (most or all the time). A higher total score indicated more severe depressive symptoms. The MC-CES-D had acceptable congruent validity (area under the receiver operative characteristic curve for major depressive disorder = 0.88–0.90), construct validity [77], internal consistency (Cronbach’s α = 0.90) and 1-week test–retest reliability (ICC = 0.93) [78]. The Cronbach’s α value for the MC-CES-D in the present study was 0.92.

#### 2.2.4. Perceived Family Support

The participants’ satisfaction with the aspects of perceived family support during childhood was evaluated using the 5-item self-reported Mandarin Chinese version [79] of the Family Adaptation, Partnership, Growth, Affection, and Resolve (APGAR) index (for example, “I am satisfied with the way my family expresses affection and responds to my feelings such as anger, sorrow, and love”) [80]. The participants rated each item on a 4-point Likert scale from 0 (never) to 3 (always). A higher total score indicated a higher level of satisfaction with perceived family support. The Cronbach’s α value for the Family APGAR index in the present study was 0.86.

#### 2.2.5. Demographic and Sexual Orientation

We collected information on the participants’ age and self-identified sexual orientation (gay or bisexual).

### 2.3. Statistical Analysis

Statistical analyses were conducted using SAS 9.4 (SAS Institute, Cary, NC, USA). Descriptive results are presented as the frequency and percentage for categorical variables and as the mean and standard deviation for continuous variables. A correlation analysis with Pearson’s *r* was performed to examine the association between the types of bullying victimization and BPD symptoms. We conducted a mediation analysis to examine whether bullying victimization directly related to BPD and depressive symptoms play a role in mediating the relationship between bullying victimization and BPD symptoms after adjusting for covariates (i.e., family support, age, and sexual orientation) and mediator and outcome variables by using the PROCESS macro (version 3.15) developed by Hayes [81] in SAS. The 95% bootstrap confidence interval (CI) with 5000 bootstrap samples was calculated to examine the statistical significance of direct and indirect effects. If the 95% bootstrap CI does not contain the null value of 0, a statistically significant indirect effect is considered.

In addition to the mediation analysis, we conducted a moderation analysis to examine the moderating role of family support in the association between bullying victimization and BPD symptoms after adjustment for covariates. The moderation analysis was conducted using a linear regression model with BPD symptoms as the outcome variable; with family support, types of bullying victimization, and the interaction between the two as predictors; and with other covariates. If the interaction term reached statistical significance (a two-sided *p* value of <0.05), family support was considered to exert a moderating effect on the relationship between bullying and BPD symptoms. The statistical concept of mediation and moderation analysis is presented in Figure 1.

## 3. Results

Table 1 summarizes the characteristics of the sample. The mean (SD) socre of BPD symptoms on the MC-BSL-23 was 18.9 (17.7). The results of the correlation analysis indicated that BPD symptoms were positively associated with the different types of bullying victimization (specifically, *r* = 0.29 for enacted stigma victimization because of gender nonconformity, *r* = 0.24 for enacted stigma victimization because of sexual orientation, *r* = 0.28 for cyberbullying victimization because of gender nonconformity, and *r* = 0.162 for cyberbullying victimization because of sexual orientation).

The results of the mediation analysis revealed that all the types of homophobic bullying victimization were positively associated with BPD symptoms because the 95% bootstrap CIs of all direct effects did not contain 0 after covariates were adjusted for (Table 2 and Figure 2). In addition, depressive symptoms positively mediated the association between homophobic bullying victimization and BPD symptoms because the 95% bootstrap CIs of all indirect effects did not contain 0.

Results are shown as the regression coefficient and 95% bootstrap confidence interval. These mediation analyses were conducted after adjustment for family support, age and sexual orientation for mediator and outcome variables.

The results of the moderation analysis revealed that family support moderated the association between homophobic bullying victimization and BPD symptoms with significant interaction effects (Table 3 and Figure 3). The significant interaction coefficients ranged from −0.09 to −0.34, indicating that the association between bullying victimization and BPD symptoms decreased when the individuals had a higher score of family support. The moderation effects appeared to be stronger for cyberbullying victimization as a result of gender nonconformity and sexual orientation than for enacted stigma victimization as a result of gender nonconformity as indicated by the difference in regression coefficients (−0.34 and −0.26 vs. −0.09). For enacted stigma victimization as a result of sexual orientation, although family support did not significantly moderate the relationship between BPD symptoms and enacted stigma victimization as a result of sexual orientation, a small *p* value of 0.07 was observed and the magnitude of the moderating effect (regression coefficient = −0.09) was similar to that of enacted stigma victimization as a result of gender nonconformity (regression coefficient = −0.09).

## 4. Discussion

The present study demonstrated that all the types of homophobic bullying victimization in childhood that were analyzed were directly associated with BDP symptoms in young adulthood among the gay and bisexual men; homophobic bullying victimization was indirectly associated with BPD symptoms through the mediation of depressive symptoms. In addition, we observed that the association between homophobic bullying victimization and BPD symptoms decreased when the individuals had more family support.

### 4.1. Homophobic Bullying Victimization and BPD Symptoms

The mean (SD) score of BPD symptoms on the MC-BSL-23 was 18.9 (17.7) among gay and bisexual men in this study. A previous study [38] assessing BPD symptoms on the MC-BSL-23 among 238 male college students in Taiwan revealed a mean (SD) score of 11.8 (10.1). Gay and bisexual men in the present study had greater BPD symptoms compared with male college students (*p* < 0.001).

The results of this study supported the hypothesis of Rodriguez-Seijas et al. [7] that bullying victimization in childhood would increase the risk of BPD symptoms later in life among LGB individuals. Bullying victimization may alter victims’ responses to stress [82] and change regulatory systems for impulse and affective control [83]. Furthermore, bullying victimization may exacerbate victims’ peer interaction problems and negatively affect their relational schemata including rejection sensitivity [84]. Moreover, according to the diathesis-stress model of BPD [85], the victims of bullying may exhibit BPD symptoms in the context of genetic vulnerability related to reactive temperament [86] and emotional regulation [87].

### 4.2. Mediation of Depressive Symptoms

The present study indicated that depressive symptoms mediated the association between homophobic bullying victimization and BPD symptoms among gay and bisexual men. Consistent with the results of previous studies [6,68,88], homophobic bullying victimization in childhood was significantly associated with depressive symptoms in adulthood. According to the minority stress theory [31], homophobic bullying is one of the most powerful stressors that marginalize LGB individuals during the development process. Homophobic bullying causes stressful experiences, anticipation of these stressful experiences, and internalized stigma in LGB individuals, all of which associate the sexual minority status with mental health problems in adulthood [89,90,91,92]. Homophobic bullying may also disturb the victims’ cognitive and emotional controls, which increases the risks of mental health problems [93].

Although the cross-sectional design of this study precluded any determination of the temporal relationship between depressive and BPD symptoms, previous studies have reported that childhood trauma can negatively affect victims’ emotion regulation mechanisms such as hypothalamic–pituitary–adrenal function [36] and predict depression [94]. Emotion regulation difficulties can increase the risk of BDP symptoms later in life [36]. On the basis of the results of this study, intervention programs for the victims of homophobic bullying should enhance their emotional regulation functions.

### 4.3. Moderation of Family Support

The present study observed that the association between homophobic bullying victimization and BPD symptoms decreased when the individuals had a higher score of family support. Family is a microsystem in which individuals are embedded; therefore, familial contexts may contribute to the formation and maintenance of personality [37]. According to the transactional model of developmental psychology [95], young individuals who perceive a protective family climate have a high degree of mature identity [96]. Family support can protect LGB individuals from mental health problems caused by stigma related to heteronormativity [97], whereas low family support may exacerbate suicidality and internalized homonegativity among young LGB individuals [98]. The results of this study indicate the importance of the family environment for LGB individuals [99,100].

### 4.4. Implications

The results of this study support the development of environment-level and individual-level interventions that can reduce homophobic bullying victimization and its negative effects on mood regulation and personality development. Regarding environmental-level interventions, intervention programs should be established in the levels of the law, society, school, and family. Implementing the law to protect LGB individuals from homophobic bullying victimization is the most fundamental strategy. Although the Taiwan government has enacted the Sexual Harassment Prevention Act, Act of Gender Equality in Employment, and Gender Equity Education Act, most people in Taiwan suppose that these laws ensure gender quality but not sexuality equality. Governments should also enhance the public’s knowledge and acceptance of various sexual orientations and reduce public stigma toward LGB individuals through the mass media. The schools are the environments in which LGB youths frequently experience homophobic bullying. Establishing the school culture of zero tolerance for homophobic bullying and enhancing peer support for LGB youth are necessary. Intervention programs designed to enhance family support should enrich relationships and communication between families and LGB individuals as well as the knowledge of family members regarding sexual orientation [101]. Regarding individual-level interventions, health professionals should routinely survey the experiences of homophobic bullying victimization among LGB individuals with depressive and BPD symptoms, helping them develop alternative cognitive and emotional coping strategies for the experiences of victimization. Affirming individuals’ sexual orientation is fundamental to improving depressive and BPD symptoms among LGB individuals. The results of previous studies in Taiwan have supported the effects of cognitive therapy [102] and dialectical behavior therapy [102,103] on reducing suicidal risk and depressive symptoms in individuals with BPD. However, whether these psychotherapy models have similar effects on BPD and depressive symptoms among gay and bisexual men experiencing homophobic bullying victimization warrants further study in Taiwan.

### 4.5. Limitations

This study has some limitations. First, because this study retrospectively explored the experiences of homophobic bullying victimization and perceived family support in childhood, recall bias might be present. Second, the cross-sectional study design limited the inference concerning the temporal relationship between depressive and BPD symptoms. It is also possible that the individuals with significant BPD symptoms might be more likely to report being a victim of childhood bullying. Third, the participants were young adult gay or bisexual men. Therefore, whether the results of this study can be generalized to lesbian and bisexual women or older individuals remain unclear. Fourth, because the data were self-reported, single-rater bias might be present. Fifth, the present study recruited participants based on their biological sex but not gender; therefore, the effect of gender minority identities could not be determined.

## 5. Conclusions

The present study demonstrated that all the types of homophobic bullying victimization during childhood that were analyzed were directly associated with BPD symptoms in young adulthood and indirectly associated with BPD symptoms through the mediation of depressive symptoms among the gay and bisexual men. Family support moderated the association between homophobic bullying victimization and BPD symptoms. Intervention programs to reduce homophobic bullying victimization and enhance family support for LGB individuals and their families are necessary. Interventions to improve depressive and BPD symptoms among LGB individuals are also necessary, especially for those who experienced homophobic bullying victimization during childhood.

## Figures and Tables

**Figure 1 ijerph-19-04789-f001:**
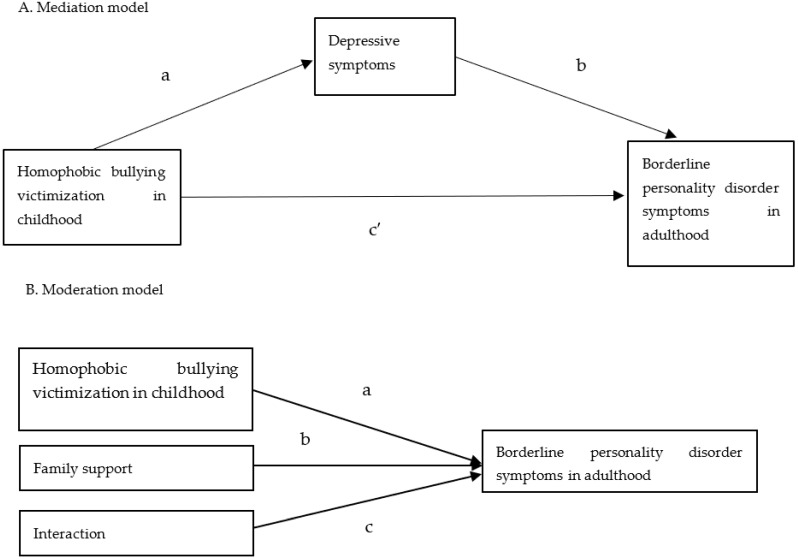
(**A**) Simple mediation and (**B**) simple moderation models depicted by statistical concepts. In a simple mediation model, a statistical mediation effect is determined by the product terms (a,b), and a mediation effect means that there is an indirect effect from the predictor on the outcome through the mediator in addition to a direct effect (c’). On the other hand, in a simple moderation model, a statistical moderation effect is determined by the interaction term (c), and a moderation effect means that the effect from the predictor on the outcome changes based on the value of the moderator dation and Moderation.

**Figure 2 ijerph-19-04789-f002:**
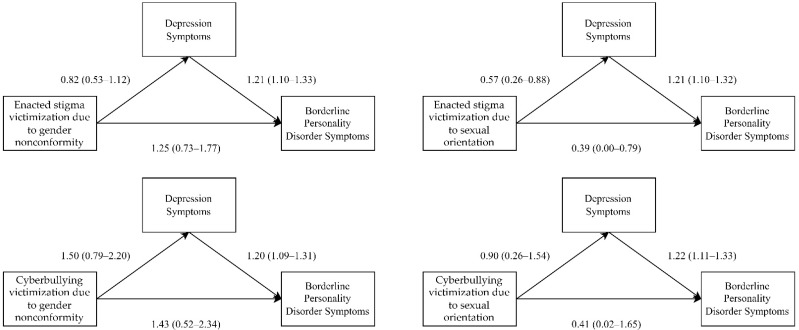
Visualization of the mediation models of Depression Symptoms on Relationship between Bullying Victimization and Borderline Personality Disorder Symptoms. Results are shown as the regression coefficient and 95% bootstrap confidence interval. These mediation analyses were conducted after adjustment for family support, age and sexual orientation for mediator and outcome variables.

**Figure 3 ijerph-19-04789-f003:**
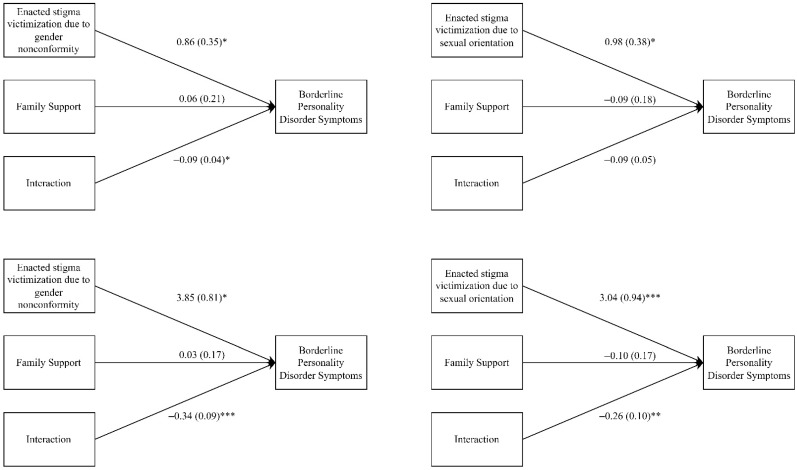
Visualization of the moderation models of Family Support in Association between Bullying Victimization and Borderline Personality Disorder Symptoms. Results are shown as the regression coefficient and standard error. These moderation analyses were conducted after adjustment for age, sexual orientation and depression. * *p* < 0.05, ** *p* < 0.01, *** *p* < 0.001.

**Table 1 ijerph-19-04789-t001:** Characteristics of Participants (N = 500).

Variable	N (%) or Mean ± Standard Deviation
Age (years)	22.9 ± 1.6
Sexual orientation	
Gay	371 (25.8)
Bisexual	129 (74.2)
Family support	8.5 ± 3.8
Bulling victimization	
Enacted stigma victimization due to gender nonconformity	3.2 ± 2.9
Enacted stigma victimization due to sexual orientation	1.8 ± 2.8
Cyberbullying victimization due to gender nonconformity	0.7 ± 1.2
Cyberbullying victimization due to sexual orientation	0.6 ± 1.3
Depression symptoms	17.5 ± 10.3
Borderline personality disorder symptoms	18.9 ± 17.7

**Table 2 ijerph-19-04789-t002:** Mediating Effect of Depression Symptoms on Relationship between Bullying Victimization and Borderline Personality Disorder Symptoms.

Mediator	Bullying Victimization → Depression	Depression → Borderline Personality Disorder Symptoms	Indirect Effect	Direct Effect Bullying Victimization → Borderline Personality Disorder Symptoms
Enacted stigma victimization due to gender nonconformity	0.82 (0.53–1.12)	1.21 (1.10–1.33)	0.73 (0.07–0.20)	1.25 (0.73–1.77)
Enacted stigma victimization due to sexual orientation	0.57 (0.26–0.88)	1.21 (1.10–1.32)	0.69 (0.25–1.18)	0.39 (0.00–0.79)
Cyberbullying victimization due to gender nonconformity	1.50 (0.79–2.20)	1.20 (1.09–1.31)	1.79 (0.77–2.89)	1.43 (0.52–2.34)
Cyberbullying victimization due to sexual orientation	0.90 (0.26–1.54)	1.22 (1.11–1.33)	1.10 (0.31–1.98)	0.41 (0.02–1.65)

**Table 3 ijerph-19-04789-t003:** Moderation of Family Support in Association between Bullying Victimization and Borderline Personality Disorder Symptoms.

Moderator	Enacted Stigma Victimization Due to Gender Nonconformity	Enacted Stigma Victimization Due to Sexual Orientation	Cyberbullying Victimization Due to Gender Nonconformity	Cyberbullying Victimization Due to Sexual Orientation
	B (SE)	B (SE)	B (SE)	B (SE)
Bullying victimization	0.86 (0.35) *	0.98 (0.38) *	3.85 (0.81) *	3.04 (0.94) ***
Family support	0.06 (0.21)	−0.09 (0.18)	0.03 (0.17)	−0.10 (0.17)
Bullying victimization by family support	−0.09 (0.04) *	−0.09 (0.05)	−0.34 (0.09) ***	−0.26 (0.10) **
Sexual orientation (gay vs. bisexual)	−0.59 (1.23)	0.36 (1.23)	0.78 (1.21)	0.46 (1.22)
Age	−0.35 (0.34)	−0.44 (0.34)	−0.36 (0.34)	−0.43 (0.34)
Depression symptoms	1.21 (0.06) ***	1.21 (0.06) ***	1.20 (0.06) ***	1.21 (0.06) ***

Abbreviations: B, unstandardized regression coefficient; SE, standard error. * *p* < 0.05, ** *p* < 0.01, *** *p* < 0.001.

## Data Availability

Anonymized data, as only described in this manuscript for analysis, are available from the corresponding author (C.-F.Y.).

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
