# Peer review of "Relationships of Homophobic Bullying Victimization during Childhood with Borderline Personality Disorder Symptoms in Early Adulthood among Gay and Bisexual Men: Mediating Effect of Depressive Symptoms and Moderating Effect of Family Support"

_ijerph, 2022, doi:10.3390/ijerph19084789_

Round 1

Reviewer 1 Report

Relationships of Homophobic Bullying Victimization During Childhood with Borderline Personality Disorder Symptoms in Early Adulthood Among Gay and Bisexual Men: Mediating Effect of Depressive Symptoms and Moderating Effect of Family Support

This is an important contribution in the field of mental health with a focus on young MSM and homophobic bullying.

Nevertheless, I have some suggestions that I would like to be seen addressed before the article is published:

  1. Line 55. This phrase should be rewritten. Etiology of BPD is not clear and reference’s 7 conclusion was drawn from clinical samples.
  2. No reference was made to the minority stress model, authors should include this model to sustain that higher prevalence of mental health problem among gay and bisexual men is due to social stress and prejudice.
  3. Authors should include information on sociodemographic measures.
  4. Also, how was gay and bisexual identity measured? How is it different from MSM?
  5. Please clarify: do all 500 participants present some form of borderline symptomatology? What does having 18.9±17.7 of borderline symptoms mean?
  6. It would be convenient to show graphic details of the moderation and mediation models.
  7. Again, results need to be discussed in the light of the minority stress theory.
  8. Authors should address implications on mental health policies and intervention.

Best wishes.

Author Response

We appreciated your valuable comments. As discussed below, we have revised our manuscript with underlines based on your suggestions. Please let us know if we need to provide anything else regarding this revision.

Comment 1

Line 55. This phrase should be rewritten. Etiology of BPD is not clear and reference’s 7 conclusion was drawn from clinical samples.

Response

Thank you for your suggestion. We rewrote this sentence as below. Please refer to line 55-58.

The negative effect of homophobic bullying victimization on personality development is an important issue for LGB individuals. Research found that LGB individuals were more likely to be diagnosed with borderline personality disorder (BPD) than heterosexual individuals [7].

Comment 2

No reference was made to the minority stress model, authors should include this model to sustain that higher prevalence of mental health problem among gay and bisexual men is due to social stress and prejudice.

Response

Thank you for your reminding. We revised the sentence describing minority stress theory and added the references as below. Please refer to line 100-105.

“The most commonly cited cause of depression in LGB individuals is “minority stress,” defined as the unique stress experienced by LGB individuals living in a social environment characterized by anti-LGB, or “heterosexist,” prejudice and stigma [31]. According to minority stress theory [31], exposure to homophobic bullying is one of the distal stressors that may leading to mental health problems such as depressive symptoms among LGB individuals [32].”

Comment 3

Authors should include information on sociodemographic measures.

Response

We included information on sociodemographic measures as below. Please refer to line 247-249.

2.2.5. Demographic and Sexual Orientation

We collected information on the participants’ age and self-identified sexual orientation (gay or bisexual).”

Comment 4

Also, how was gay and bisexual identity measured? How is it different from MSM?

Response

In this study, gay and bisexual identity were self-identified. Compared with the identity of men who have sex with men (MSM), the self-identifies of gay and bisexuality could reduce participants’ uncertainty about the definition of “having sex with men” and resistance to participate into this study.

Comment 5

Please clarify: do all 500 participants present some form of borderline symptomatology? What does having 18.9±17.7 of borderline symptoms mean?

Response

Thank you for your comment. In the revised manuscript we compared the participants’ mean score of BPD symptoms in this study with that of a previous study using the same instrument to assess BPD symptoms among male college students in Taiwan. The result indicated that gay and bisexual men in the present study had greater BPD symptoms compared with male college students. We added the content as below. Please refer to line 346-350.

The mean (SD) score of BPD symptoms on the MC-BSL-23 was 18.9 (17.7) among gay and bisexual men in this study. A previous study [38] assessing BPD symptoms on the MC-BSL-23 among 238 male college students in Taiwan revealed a mean (SD) score of 11.8 (10.1). Gay and bisexual men in the present study had greater BPD symptoms compared with male college students (p < 0.001).

Comment 6

It would be convenient to show graphic details of the moderation and mediation models.

Response

Thank you for your suggestion. We added the graphic details of the moderation and mediation models into the revised manuscript. Please refer to Figures 1 to 3.

Comment 7

Again, results need to be discussed in the light of the minority stress theory.

Response

Thank you for your suggestion. We revised the content of Discussion in the light of the minority stress theory as below. Please refer to line 363-371.

Consistent with the results of previous studies [6,68,88], homophobic bullying victimization in childhood was significantly associated with depressive symptoms in adulthood. According to the minority stress theory [31], homophobic bullying is one of the most powerful stressors that marginalize LGB individuals during the development process. Homophobic bullying causes stressful experiences, anticipation of these stressful experiences, and internalized stigma in LGB individuals, all of which associate the sexual minority status with mental health problems in adulthood [89-92]. Homophobic bullying may also disturb the victims’ cognitive and emotional controls, which increases the risks of mental health problems [93].

Comment 8

Authors should address implications on mental health policies and intervention.

Response

Thank you for your suggestion. We rewrote the content of 4.4. Implements as below to address multiple levels of intervention (including mental health). Please refer to line 392-418.

The results of this study support the development of environment-level and individual-level interventions that can reduce homophobic bullying victimization and its negative effects on mood regulation and personality development. Regarding environmental-level interventions, intervention programs should be established in the levels of the law, society, school, and family. Implementing the law to protect LGB individuals from homophobic bullying victimization is the most fundamental strategy. Although the Taiwan government has enacted the Sexual Harassment Prevention Act, Act of Gender Equality in Employment, and Gender Equity Education Act, most people in Taiwan suppose that these laws ensure gender quality but not sexuality equality. Governments should also enhance the public’s knowledge and acceptance of various sexual orientations and reduce public stigma toward LGB individuals through the mass media. The schools are the environments in which LGB youths frequently experience homophobic bullying. Establishing the school culture of zero tolerance for homophobic bullying and enhancing peer support for LGB youth are necessary. Intervention programs designed to enhance family support should enrich relationships and communication between families and LGB individuals as well as the knowledge of family members regarding sexual orientation [101]. Regarding individual-level interventions, health professionals should routinely survey the experiences of homophobic bullying victimization among LGB individuals with depressive and BPD symptoms, helping them develop alternative cognitive and emotional coping strategies for the experiences of victimization. Affirming individuals’ sexual orientation is fundamental to improving depressive and BPD symptoms among LGB individuals. The results of previous studies in Taiwan have supported the effects of cognitive therapy [102] and dialectical behavior therapy [102,103] on reducing suicidal risk and depressive symptoms in individuals with BPD. However, whether these psychotherapy models have similar effects on BPD and depressive symptoms among gay and bisexual men experiencing homophobic bullying victimization warrants further study in Taiwan.

Reviewer 2 Report

Thank you for the opportunity to review this article. This article provides novel and crucial findings on the mediating role of depression symptoms and moderating role of family support on the relationship between childhood bullying victimisation and prevalence of borderline personality disorder. There are several strengths of the manuscript, such that it is well-written, the descriptions of the result and discussion sections are clear, and the implication section provides useful indicators for change. See below for my suggestions to the authors:

  1. In some countries, using the “minority” terminology is beginning to be seen as perpetuating stigma of being a minority. I wonder if “sexual minority” is a common and appropriate term to refer to Taiwanese gay and bisexual people?
  2. The authors have done an incredible job in summarising the international evidence on the prevalence and correlates of borderline personality disorder. However, readers are also interested in the specific cultural context of Taiwan. Can the authors a) expand on the rationale or importance of conducting this study in Taiwan? b) include a paragraph that reviews existing studies on the mental health status of Taiwanese gay and bisexual people to set the context for this study?
  3. Line 58 – Can the authors clarify if the reported prevalence of BPD was from US studies?
  4. Line 69 – Being gay or bisexual is an “identity” rather than a “status”. Perhaps the author can restructure the sentence to “Studies have reported many of those diagnosed as having BPD identified as gay or bisexual.” The authors can also consider having an example of the prevalence of BPD among gay and bisexual populations to demonstrate the severity of this issue.
  5. Line 116 – Did this sample of men included transgender men? Otherwise, the authors can consider using the language “cisgender men”.
  6. Line 133 – The term “traditional bullying” is not common in public health literature. Have the authors considered using the language of “enacted stigma” or “overt discrimination” as a substitute? Here are some references that the authors can tap into:

Poon, C., Saewyc, E., & Chen, W. (2011). Enacted stigma, problem substance use, and protective factors among Asian sexual minority youth in British Columbia. Canadian Journal of Community Mental Health, 30(2), 47–64. doi:10.7870/cjcmh-2011–0016

O’Connor, L. K., Pleskach, P., & Yanos, P. (2018). The experience of dual stigma and self- stigma among LGBTQ individuals with severe mental illness. American Journal of Psychiatric Rehabilitation, 21(1), 167–187 https://www.muse.jhu.edu/article/759951

  1. Line 142 – I struggle to connect the dots between “gender nonconformity” and the experiences of being gay and bisexual that have little relevance to gender. Associating gay and bisexual identities with “sissy” attribute is a form of stereotype that does not hold true for many within the communities. Can the authors elaborate on the rationale of examining “gender nonconformity” – a construct that is more relevant to the transgender communities?
  2. I could not locate the scale with the elements of “gender nonconformity” and “sexual orientation” from the reference cited (34) : https://onlinelibrary.wiley.com/action/downloadSupplement?doi=10.1016%2Fj.kjms.2012.04.008&file=kjm2500-sup-0001.pdf

If the authors adapted the MC-SBEQ scale for gay and bisexual people, the process of doing so should be made clear to the readers for the purpose of replicability.

  1. Table 1 – I suggest bolding the variable name (age, sexual orientation, and bullying victimisation).
  2. Line 302 – The term “homonegative” stigma should be replaced by “stigma related to heteronormativity” to also encompass the prejudicial experiences affecting bisexual individuals.
  3. The implications section can be further enhanced by incorporating the specific cultural context of Taiwan. For example, what are the existing interventions available to BPD patients in Taiwan? Do existing BPD interventions provide culturally-competent services to gay and bisexual people? What are the existing policies and laws available to protect gay and bisexual people in Taiwan from enacted stigma/overt discrimination?
  4. Section 4.5. Some reviewers can be far too harsh with regards to the use of mediation analysis in cross sectional data. I think that this approach can be justified and the authors did a good job of engaging with its limits. That said, while I think childhood bullying victimisation would indeed be likely to have had temporal precedence on BPD, the reverse association cannot be ruled out (i.e., BPD patients are more likely to report being a victim of childhood bullying). I recommend the author to note this as a limitation and urge for future studies to undertake a longitudinal analysis.

Author Response

We appreciated your valuable comments. As discussed below, we have revised our manuscript with underlines based on your suggestions. Please let us know if we need to provide anything else regarding this revision.

Comment 1

In some countries, using the “minority” terminology is beginning to be seen as perpetuating stigma of being a minority. I wonder if “sexual minority” is a common and appropriate term to refer to Taiwanese gay and bisexual people?

Response

Thank you for your comment. We replaced “sexual minority individuals” by “Lesbian, gay and bisexual individuals” thorough the revised manuscript.

Comment 2

The authors have done an incredible job in summarising the international evidence on the prevalence and correlates of borderline personality disorder. However, readers are also interested in the specific cultural context of Taiwan. Can the authors a) expand on the rationale or importance of conducting this study in Taiwan? b) include a paragraph that reviews existing studies on the mental health status of Taiwanese gay and bisexual people to set the context for this study?

Response

Thank you for your suggestion. We added a new paragraph to introduce rewrote this sentence as below. Please refer to line 123-143.

“1.4. Mental Health among LGB Individuals in Taiwan

The attitude toward homosexuality has even been supposed to be socially tolerable among Taiwanese society in the past two decades [41]. The Taiwan government has enacted the Sexual Harassment Prevention Act [42], Act of Gender Equality in Employment [43], and Gender Equity Education Act [44] as well as legislated the same-sex relationship [45] to protect the rights of LGB individuals from being violated. However, people in Taiwan traditionally regard homosexuality as a challenge to the family obligations mandated in Confucianism, and in particular, they require their offspring to continue the family bloodline [41]. Gay and bisexual men are still seen as the symbol of disseminating human immunodeficiency virus infection and deteriorating traditional family values [46]. Victimization of homophobic bullying [47,48], public stigma [49], microaggression [50,51], and internalized stigma [52] are prevalent among LGB individuals. Especially, the groups opposing same-sex marriage in Taiwan spread a considerable amount of incorrect information and rumors to malign LGB individuals through social media and public media; these misleading portrayals and negative stereotypes spread in the media demoralized LGB individuals and directly disturbed their emotional regulation [53-58]. LGB individuals have higher rates of suicidality [59,60], addictive substances use [61,62], problematic internet and smartphone use [63], poor quality of life [64,65], depression [66], and psychological inflexibility [67] compared with heterosexual ones in Taiwan. Therefore, mental health is an important issue warranted further study among LGB individuals in Taiwan.

Comment 3

Line 58 – Can the authors clarify if the reported prevalence of BPD was from US studies?

Response

We revised the contents regarding the prevalence of BPD as below made the locations of studies clearer. Please refer to line 59-63.

In epidemiological studies of adults in the USA, prevalances for BPD were between 0.5% and 5.9% in the general US population [8,9] with a median prevalence of 1.35% [10]. Previous studies around the world, most conducted in the USA and Europe, found that in clinical populations, BPD is the most common personality disorder, with a prevalence of 10% of all psychiatric outpatients and between 15% and 25% of inpatients [11,12].

Comment 4

Line 69 – Being gay or bisexual is an “identity” rather than a “status”. Perhaps the author can restructure the sentence to “Studies have reported many of those diagnosed as having BPD identified as gay or bisexual.” The authors can also consider having an example of the prevalence of BPD among gay and bisexual populations to demonstrate the severity of this issue.

Response

Thank you for your suggestion. We revised the sentence based on your suggestion. We also added the examples for the association between the diagnosis of BPD and sexual orientation as below. Please refer to line 73-80.

Studies have reported many of those diagnosed as having BPD identified as gay or bisexual [24,25]. For example, a clinical study found that homosexuality was ten times more common in men and six times more common in women with BPD compared to both the general population and a depressed control group [26]. A ten-year longitudinal study on a clinical sample found that approximately one-third of both men and women with BPD engaged in homosexual relationships [27]. A study on the individuals referred for psychiatric treatment revealed that LGB individuals were more likely to be diagnosed with BPD than heterosexual individuals (odds ratio = 2.43, p < 0.001) [7].

Comment 5

Line 116 – Did this sample of men included transgender men? Otherwise, the authors can consider using the language “cisgender men”.

Response

We recruited participants based on their biological sex but not gender; therefore, the effect of gender minority identities could not be determined. We have listed it as one of the limitations of this study. Please refer to line 428-429.

“Fifth, the present study recruited participants based on their biological sex but not gender; therefore, the effect of gender minority identities could not be determined.”

Comment 6

Line 133 – The term “traditional bullying” is not common in public health literature. Have the authors considered using the language of “enacted stigma” or “overt discrimination” as a substitute? Here are some references that the authors can tap into:

Poon, C., Saewyc, E., & Chen, W. (2011). Enacted stigma, problem substance use, and protective factors among Asian sexual minority youth in British Columbia. Canadian Journal of Community Mental Health, 30(2), 47–64. doi:10.7870/cjcmh-2011–0016

O’Connor, L. K., Pleskach, P., & Yanos, P. (2018). The experience of dual stigma and self- stigma among LGBTQ individuals with severe mental illness. American Journal of Psychiatric Rehabilitation, 21(1), 167–187 https://www.muse.jhu.edu/article/759951

Response

Thank you for your suggestion. We used “enacted stigma” to replace “traditional bullying” thorough the revised manuscript and cited “Poon et al. (2011)” as the reference (reference 69).

Comment 7

Line 142 – I struggle to connect the dots between “gender nonconformity” and the experiences of being gay and bisexual that have little relevance to gender. Associating gay and bisexual identities with “sissy” attribute is a form of stereotype that does not hold true for many within the communities. Can the authors elaborate on the rationale of examining “gender nonconformity” – a construct that is more relevant to the transgender communities?

Response

Thank you for your comment. We added the rationale to consider gender nonconformity in surveying bullying victimization among LGB individuals as below. Please refer to line 90-94.

A literature review also reported that gender role nonconformity significantly increases the risk of experiencing sexuality-related bullying in both heterosexual and LGB populations [29]. Therefore, the experiences of both homophobic bullying victimization as a result of sexual orientation and gender nonconformity should be considered.

Reference 29. Reference: Hong, J.S.; Garbarino, J. Risk and protective factors for homophobic bullying in schools: An application of the social–ecological framework. Educ. Psychol. Rev. 2012, 24, 271–285, doi:10.1007/s10648-012-9194-y.”

Comment 8

I could not locate the scale with the elements of “gender nonconformity” and “sexual orientation” from the reference cited (34): https://onlinelibrary.wiley.com/action/downloadSupplement?doi=10.1016%2Fj.kjms.2012.04.008&file=kjm2500-sup-0001.pdf

If the authors adapted the MC-SBEQ scale for gay and bisexual people, the process of doing so should be made clear to the readers for the purpose of replicability.

Response

Thank you for your reminding. We added the description for adapting the MC-SBEQ scale for gay and bisexual people as below. Please refer to line 190-193.

A previous study adapting the MC-SBEQ for measuring the victimization of bullying due to gender nonconformity and sexual orientation in gay and bisexual men and found the total McDonald’s ω values of were 0.85 and 0.92, respectively, indicating a acceptable internal consistency [47]

Reference 47: Wang, C.C.; Hsiao, R.C.; Yen, C.F. Victimization of traditional and cyber bullying during childhood and their correlates among adult gay and bisexual men in Taiwan: A retrospective study. Int. J. Environ. Res. Public Health 2019, 16, 4634. doi: 10.3390/ijerph16234634.”

Comment 9

Table 1 – I suggest bolding the variable name (age, sexual orientation, and bullying victimisation).

Response

Thank you for your suggestion. We revised the variable names in Table 1 to make them easily read. Please refer to Table 1.

Comment 10

Line 302 – The term “homonegative” stigma should be replaced by “stigma related to heteronormativity” to also encompass the prejudicial experiences affecting bisexual individuals.

Response

Thank you for your suggestion. We replaced the term based on your suggestion as below. Please refer to line 387.

“Family support can protect LGB individuals from mental health problems caused by stigma related to heteronormativity…”

Comment 11

The implications section can be further enhanced by incorporating the specific cultural context of Taiwan. For example, what are the existing interventions available to BPD patients in Taiwan? Do existing BPD interventions provide culturally-competent services to gay and bisexual people? What are the existing policies and laws available to protect gay and bisexual people in Taiwan from enacted stigma/overt discrimination?

Response

Thank you for your suggestion. As mentioned in the response to Comment 2, we added the new paragraph (1.4. Mental Health among LGB Individuals in Taiwan) to introduce the laws available to protect gay and bisexual people in Taiwan from enacted stigma. We also rewrote the content of 4.4. Implements as below to introduce the results of previous studies examining the effects of cognitive therapy and dialectical behavior therapy among individuals with BPD in Taiwan. Please refer to line 392-418.

The results of this study support the development of environment-level and individual-level interventions that can reduce homophobic bullying victimization and its negative effects on mood regulation and personality development. Regarding environmental-level interventions, intervention programs should be established in the levels of the law, society, school, and family. Implementing the law to protect LGB individuals from homophobic bullying victimization is the most fundamental strategy. Although the Taiwan government has enacted the Sexual Harassment Prevention Act, Act of Gender Equality in Employment, and Gender Equity Education Act, most people in Taiwan suppose that these laws ensure gender quality but not sexuality equality. Governments should also enhance the public’s knowledge and acceptance of various sexual orientations and reduce public stigma toward LGB individuals through the mass media. The schools are the environments in which LGB youths frequently experience homophobic bullying. Establishing the school culture of zero tolerance for homophobic bullying and enhancing peer support for LGB youth are necessary. Intervention programs designed to enhance family support should enrich relationships and communication between families and LGB individuals as well as the knowledge of family members regarding sexual orientation [101]. Regarding individual-level interventions, health professionals should routinely survey the experiences of homophobic bullying victimization among LGB individuals with depressive and BPD symptoms, helping them develop alternative cognitive and emotional coping strategies for the experiences of victimization. Affirming individuals’ sexual orientation is fundamental to improving depressive and BPD symptoms among LGB individuals. The results of previous studies in Taiwan have supported the effects of cognitive therapy [102] and dialectical behavior therapy [102,103] on reducing suicidal risk and depressive symptoms in individuals with BPD. However, whether these psychotherapy models have similar effects on BPD and depressive symptoms among gay and bisexual men experiencing homophobic bullying victimization warrants further study in Taiwan.

Comment 12

Section 4.5. Some reviewers can be far too harsh with regards to the use of mediation analysis in cross sectional data. I think that this approach can be justified and the authors did a good job of engaging with its limits. That said, while I think childhood bullying victimisation would indeed be likely to have had temporal precedence on BPD, the reverse association cannot be ruled out (i.e., BPD patients are more likely to report being a victim of childhood bullying). I recommend the author to note this as a limitation and urge for future studies to undertake a longitudinal analysis.

Response

Thank you for your reminding. We added it as one of the limitations of this study as below. Please refer to line 424-425.

“It is also possible that the individuals with significant BPD symptoms might be more likely to report being a victim of childhood bullying.”

Round 2

Reviewer 1 Report

Thank you for implementing all the requested changes; it has increased the overall quality of the manuscript and is now fit for publication. 

Best wishes.